# Telerehabilitation Pathways in Specific Learning Disorders: Improving Reading and Writing

**DOI:** 10.3390/brainsci13030479

**Published:** 2023-03-11

**Authors:** Agnese Capodieci, Daniela Graziani, Valentina Scali, Susanna Giaccherini, Luciano Luccherino, Chiara Pecini

**Affiliations:** 1Department of Education, Languages, Intercultures, Literatures and Psychology, University of Florence, 50121 Florence, Italy; 2Azienda USL Toscana Sudest, 52100 Arezzo, Italy

**Keywords:** telerehabilitation, neurodevelopmental disorders, specific learning disorder, reading speed, reading accuracy, writing accuracy, dictation, executive functions, interference control, cognitive flexibility

## Abstract

Telerehabilitation has proved to be a useful tool for neurodevelopmental disorders in allowing timely and intensive intervention and preventing relapses; it is also widely used for specific learning disabilities (SLD), showing significant effects on reading abilities, but variables linked to its effectiveness have not been studied yet. The present study was aimed at testing the effectiveness of telerehabilitation on reading and writing in SLD children, comparing different treatment pathways, and considering the impact of training intensity and executive functions. Seventy-three children were enrolled (telerehabilitation group: 48 children, waiting list group: 25 children). The results showed significant improvements in reading fluency, text dictation, and executive functions in the training group. Children attending a combined training including reading tasks and rapid automatized naming processes improved in word reading fluency and text dictation. The number of training sessions and the change in executive functions significantly correlated with changes in reading accuracy. Here we show a new contribution to telerehabilitation research in SLD: telerehabilitation significantly enhanced learning abilities and executive functions. Training based on the learning task and the underlying processes significantly increased not only reading speed, according to previous studies, but also writing accuracy. The findings’ implications in clinical research and practice are discussed.

## 1. Introduction

Over the past few years, long before the COVID-19 pandemic, the advent of digital healthcare has given rise to telemedicine. In accordance with the WHO (World Health Organization) [1], telemedicine is the administration of care services using technologies, to generate a beneficial effect on the health of the citizens [2]. Technologies, used daily by many people worldwide (such as computers, tablets, or smartphones), are becoming an important component in the field of digital health [3] in different disciplines and with different ages [4].

Recent years have emphasized the need to intervene in neurodevelopmental disorders with early and intensive treatments to counter the “waterfall effect” of a problem in other areas of development [5,6,7]. The long waiting lists and high costs of rehabilitation often make treatment non-timely and excessively prolonged over time, making the therapeutic pathway less effective and causing long-term negative repercussions on psychological, social, and educational opportunities and contributing to the structuring of psychological disorders [6,8,9,10]. Among the neurodevelopmental disorders, the ones that are most demanding in public service in terms of resources are specific learning disabilities (SLD). 

SLD are defined as neurodevelopmental disorders in which children have difficulties in critical academic skills, i.e., reading, written expression, and/or mathematics, for at least 6 months; these difficulties are not due to a low IQ, other sensory or neurological problems, or poor educational opportunities [11]. SLD impact roughly 5–10% of school-aged children, making them one of the most common neurodevelopmental disorders. SLD are caused by genetic and neurobiological factors that modify brain function by affecting one or more cognitive processes connected to literacy. Considering evidence from pathogenesis and biochemical hallmarks, studies have shown mitochondrial impairment in neurodevelopmental disorders [12,13] but only one study considered learning disabilities [14]. Concerning evidence from neuroimaging across perception, memory, and academic domains, neurodevelopmental deficits in different cognitive areas share a reduced long-range projection between the local perceptual areas that govern consciousness and the high-level prefrontal cortex [15]. However, SLD remain a complex disease, and indeed, the current literature analyses show that new gene clusters are constantly emerging, and their protein products are involved in numerous biological processes. Associations with SLD have been revealed by changes in brain anatomy, connectivity, and function (neuroimaging studies) or disrupted cellular mechanisms (functional studies), and whether these associations are genuine [16]. It raises the need for further research to give confidence that they are. 

Specific learning disorders may be encountered separately or in comorbidity. The comorbidity between learning disorders is four to five times higher in samples that have already experienced marked problems in one academic domain compared to the full population [17] and high comorbidities exist even with other neurodevelopmental and emotional or behavior disorders (i.e., attention deficit hyperactivity disorder, conduct disorder, anxiety, depression [18,19]). The variability of multimodal associative cortical connectivity may reflect a common origin in seemingly disparate neurodevelopmental disorders: this framework helps describe the comorbidities of neurodevelopmental disorders [15].

Analyzing specifically, children with reading disabilities experience significant impairment in the acquisition of fluidity and accuracy in reading, in the automation of the mechanism of grapheme–phoneme conversion, and in the access to the lexicon, and sometimes also encounter problems in the understanding of the written text. 

The literature on reading disabilities shows that the degree of modifiability of the process of grapheme–phoneme conversion and vice versa is still an object of widening randomized studies on dyslexia to show that different therapeutic approaches produce effects, even if of modest or little entity. A recent meta-analysis of 22 randomized controlled studies on dyslexia and dysorthography [20] showed that phonetic–phonologic instruction is not only the most frequently studied approach, but, also, the only approach whose effectiveness in improving the performances of reading and orthography in children and adolescents with reading disabilities has been confirmed through statistical analysis of the pre- and post-intervention effects, while the middle dimensions of the effect of the remaining approaches have not reached statistical significance. The meta-analysis demonstrates, in line with previous results, that important difficulties in reading and orthography can improve with appropriate treatment. The authors of the meta-analysis [20], valuing the utilization of this methodology not only in opaque languages (English) but also in those with transparent orthography such as Italian, conclude that the systematic teaching of grapheme–phoneme correspondence and strategies of decoding, the work on the phonologic consciousness, and the application of these abilities in the activities of reading and writing are the most effective methods to improve the abilities in the diffusion of literacy skills of children and adolescents with these difficulties. 

From the analysis of the literature, as far as it regards reading disabilities, as well as the meta-analysis of Galuschka and colleagues [20], another 23 randomized control studies emerge [21], from which it emerges that the treatment that shows a certain degree of effectiveness (effect size between 0.40 and 0.60) in improving reading accuracy and speed, is explicit training in the use of sublexical transcoding strategies (association between grapheme and phoneme) through reading and writing tasks that require the merging or segmentation of strings of letters into single graphemes or phonemes, into syllables or into rhymes. It is also effective to combine phonological–meta-phonological interventions with “multi-component” interventions, aimed at enhancing transcoding, lexical competence, morphosyntactic awareness, and strategies useful for understanding passages, to induce improvements both in the speed and correctness of reading and in understanding the text.

Children with writing disorders experience a significant impairment in phoneme–graphemic conversion, producing texts with a high number of errors. Regarding dysorthography, in addition to the meta-analysis by Galuschka and colleagues [20], a systematic review [22] and two randomized controlled trials [23,24] were found. From these studies emerge as an effective treatment, in the first cycle of primary school, focusing on enhancing the phoneme–grapheme transcription processes. From the second cycle of primary school, the literature suggests that “multi-component” interventions, not only oriented towards orthographic competence (morphology and structure of words) and the representation of orthographic patterns but also towards strengthening executive functions (working memory and inhibition of answer), result in being more effective.

Considering what has emerged from the literature, the importance of training on grapheme–phoneme conversion processes and vice versa is clear, together with training on more general cognitive processes such as executive functions. In fact, the latter underlies learning abilities and can allow greater compensation for children with learning difficulties. Specific training focused on executive functions, according to the Diamond model [25], showed significant enhancements in verbal fluency, rapid naming speed, reading accuracy and speed, and writing accuracy, and significant correlations among inhibition speed and accuracy, and writing accuracy [26].

Being able to propose training that combines these aspects through a classic face-to-face intervention that includes one meeting a week, or a maximum of two, with the clinician is complicated. The use of platforms and applications, specifically created to train reading and writing (grapheme–phoneme conversion and vice versa, meta-phonological awareness, etc.) and the general processes that underlie them (executive functions), allow a more intense practice to be carried out using the exercises for a few minutes several times a week.

In recent years, many tools for telerehabilitation have spread for the treatment of SLD. Thanks to the development of increasingly advanced technologies, innovative tele-rehabilitative techniques have progressively been disseminated that allow the execution of specific exercises through remote monitoring by the clinician; in this particular case, they consist of intensive and self-adaptive activities, in which the difficulty and duration of the task change automatically, adapting to the performance of the child, so that the complexity of the exercises is challenging [10,27]. These characteristics, together with the pleasantness and attractiveness, and the possibility to offer simultaneously multiple and multimodal stimulations of more sensory channels and immediate feedback on the accuracy and the level achieved through the software itself, are the basis of the greater benefits often documented in these types of interventions compared with traditional ones [28,29,30]. In addition, telerehabilitation offers many potential benefits for both the patient and the professional. For the user, it allows them to spend less time and money, as families can enjoy intensive intervention, as necessary for SLD, without taking the child to the clinic several times a week and without having to apply for work permits to accompany them. It is also less time-consuming for professionals because, thanks to the work of patients at home, it allows more of the work to be taken over by encouraging monitoring activities and the adoption of correct self-care practices. This is crucial for a public service where the long waiting times for traditional therapy make the intervention not timely and sometimes inaccessible. As reported by both parents and children, technology interventions are effective tools for improving children’s learning because they reflect the digitization of many educational settings and increase motivation and engagement [27]. Despite the many advantages, there is still partial resistance to telerehabilitation for children with SLD. One reason is that clinical and educational contexts are usually based on face-to-face interactions and the lack of these direct interactions may in some cases limit the effectiveness of interventions. This can be managed through initial, final, and monitoring meetings in person or through video call platforms. Moreover, studies show that not all children find telerehabilitation effective [31], and the variables connected to the inefficacy of the training, i.e., previous language disorder, diagnostic variables, and executive function profile, have not been analyzed yet.

The existing literature, although still limited, shows the validity of telerehabilitation in children with SLD. A recent review by Ogourtsova and colleagues [7] included 55 studies from 6 diagnostic groups and 10 health professionals. Common telerehabilitation goals varied by diagnostic group and included motor function, behavior, language, and parental self-efficacy. Telerehabilitation was found to be more effective or comparable to comparative interventions in improving outcomes. There is ample evidence that telerehabilitation is a promising alternative when in-person care is limited [32]. It is comparable to the usual care and more effective than no treatment. A combination of face-to-face and remote rehabilitation approaches may be beneficial for the future of post-pandemic pediatric rehabilitation. As regards efficacy studies, in a multicenter study on the treatment of dyslexia [33], the outcomes of five different treatments for the improvement in the speed and correctness of reading were compared from five different rehabilitation centers, comparing outpatient and home delivery. This study shows that a better reading speed is possible than expected by a natural evolution (about 0.3 syllables per second per school year [34]) for most participants, and confirmation is shown that at equal hours per month of treatment, equivalent changes can be achieved by both outpatient and home treatment [33]. In the Italian context, the most used platform is RIDInet. It is a platform where apps for learning and language disorders are inserted, conceived by a scientific committee based on scientific literature. The platform ensures constant supervision and monitoring by the clinician, allowing a continuous adaptation of the intervention to the results. The apps help patients to also generalize goals to non-therapeutic contexts and guarantee the continuity of the intervention, promoting the capacity for self-correction and motivation.

In the literature, several studies have used the RIDInet platform for intervention and enhancement in SLD, and they have shown interesting efficacy results in the use of apps. Most of the efficacy studies using the RIDInet platform focused on reading disorder intervention and highlighted the effectiveness of the Reading Trainer 2 app for improving reading performance, both using the single app and in conjunction with others [31]. Reading Trainer 2 is an app that allows children to speed up and makes text reading more accurate. The first results regarding the effectiveness of this type of telerehabilitation treatment demonstrate that the accuracy of loud decoding and the rate of decoding, significantly improve after about three months of treatment, regardless of the anamnestic history or neurofunctional profile of children [10,35]. Several parameters have been considered in the research, such as the use of the syllabic unit rather than the word, the calibrated starting speed on the individual or the sample, and the use of the manual mode rather than timed, and it has emerged how, even after changing these variables, it turns out that the intervention with the app offers good effectiveness [10,31,35,36].

Another application of the RIDInet platform, called Run the Ran, aims to promote the enhancement of rapid naming processes (RAN), considered important for the acquisition of reading, both in typical development and in dyslexia. The exercise aims to further increase the fundamental visual–verbal integration processes for encoding a written text [37]. This type of rehabilitation is a form of “process intervention,” as it is aimed at enhancing cognitive skills that support defective learning competence. Pecini and colleagues [38] compared two groups of children with reading disorders compared on age, sex, IQ, and reading speed. One group performed with Reading Trainer 2 (n = 21), while the other group performed with Run the Ran (n = 24); both interventions lasted 3 months. Both interventions showed significant improvements in reading speed and accuracy in naming. The enhancement through quick naming exercises allows bypassing the use of alphanumeric stimuli by enhancing the cognitive processes at the base of the reading.

Other studies have focused on other aspects of learning, such as spelling skills [39], text comprehension skills [27], calculation skills [40], rapid automatized naming [38], and working memory [26].

In light of the results of the literature, a gap has emerged in telerehabilitation with respect to training that allows for an improvement in terms of both scholastic skills and the executive functions that underlie them and that allows for compensating for the difficulties in children with DSA. For SLD, the main areas of difficulty are usually decoding and writing words. These difficulties can cause problems in other academic areas such as math, science, and social studies. As a result of this, the first objective of this study is to verify the effectiveness of telerehabilitation using RIDInet platform apps on reading and writing abilities, comparing children who attend the training with children on the waiting list. Given the association between reading abilities and the RAN process [38], a subgroup of children that used an app on the reading abilities (Reading Trainer 2) associated with one on the RAN (Run the Ran) was compared with the group on the waiting list to analyze how this association can influence the effectiveness of the training. Subsequently, correlations between the number of sessions and changes in learning abilities were conducted to consider if less or more intense training could influence the results. Finally, the performances of executive functions evaluated through remote assessment were analyzed to test if the training can also influence the executive functions’ performances, and the correlations between changes in the executive functions’ performances and in learning abilities were considered, to focus on how executive functions and learning abilities may influence each other.

## 2. Methods

An ABA design (pre-test, treatment, post-test) was applied to verify the efficacy of the training [41]. Each participant completed the pre-test assessment consisting of two face-to-face sessions at the specialist’s clinic in which parents were asked to fill out an anamnestic questionnaire and the children were given tests for the assessment of learning, and a remote session for the tele-evaluation of executive functions. The study was conducted in accordance with the recommendations of the ethics committee of the University of Florence and approved by our institutional committee, and parents’ informed written consent was obtained for all participants.

### 2.1. Participants

The study involved a convenience sample of 73 children (mean age in months = 9.34, SD = 1.42, 35 males) followed by the public health service for learning disabilities. The children attended the following school classes: 1 second grade, 29 third grade, 30 fourth grade, and 13 fifth grade. All the children carried out an initial cognitive and learning assessment so that the clinicians could have a diagnostic hypothesis. Most of them showed a mixed learning disorder (75%). The number of children who underwent the telerehabilitation intervention was 48 with an average age of 9.38 (1.16) and 22 males and 26 females. The number of children in the waiting list group was 25 with mean ages of 9.28 (0.89) and 13 boys and 12 girls. The two groups were comparable in terms of age (*F*(1.71) = 0.06, *p* > 0.05) and gender (χ^2^ = 0.25, *p* > 0.05).

It was established as exclusion criteria a history of neurological, psychiatric, or other serious psychological problems and comorbidity with other neurodevelopmental disorders. Thus, no children with SLD had comorbidity with attention and hyperactivity disorders (Conners scale ADHD index: all subjects had a score below or equal to T = 65, M = 12.98 (8.46); [42]). Given the choice of apps for the intervention protocol, which concerned the literacy domain, children who had a primary diagnosis of math disorder were excluded. Children who did not complete the 3 months of telerehabilitation or who used the apps less than twice a week were excluded from the sample.

The inclusion criteria were diagnosis of learning disability including reading and writing areas (reading fluency z score ≤ −1.5 SD, reading, and writing accuracy score ≤ 5° percentile rank) and IQ higher than 80; the children with SLD performed the WISC-IV test during the diagnostic evaluation process. All children had intelligence within the normal range (IQ > 80 at the WISC-IV with M = 96.85, DS = 12.59 [43]). The children were not receiving any other treatment.

### 2.2. Procedure

The assessment was carried out in three sessions: two in person and one remotely. In the face-to-face sessions, the clinicians carried out the cognitive and learning assessment. If children met the inclusion criteria, the clinician proposed them and their parents to be involved in the study after they expressed the informed consent. After the face-to-face assessment, in the remote session, a professional assessed the executive functions through the TeleFE platform (for more information see Rivella et al. [44]). All the children carried out the initial assessment; subsequently, 48 children undertook a telerehabilitation program with the RIDInet platform, while 25 remained on the waiting list. After the initial assessment, for children starting the intervention, a meeting was scheduled with the clinician who explained to the child and the family how the platform and app function. The choice of apps was made based on the profile that emerged from the initial assessment, but, for all children, the first app had the objective of enhancing the learning ability where they showed more difficulty, and the second app was related to more basic processes (rapid automated naming and working memory). After the meeting, the platform was activated for a 10-day trial, which allowed the professional to assess the family’s compliance with the intervention to be performed at home. If the child exercised at least 4 times in the 10 days, the apps were activated. Connection to the RIDInet website was required to access the app, two or three times a week for each app and more or less 15/20 min. The period of use was between 3 and 4 months (for the number of sessions see results). During the 3 months of telerehabilitation, the professional met the child every three weeks for face-to-face monitoring. After this period, the children who had completed the telerehabilitation period with the apps carried out a post-treatment evaluation; meanwhile, the children on the waiting list carried out a new assessment of their learning before starting the telerehabilitation process with the RIDInet platform.

### 2.3. Measures

#### In-Person Assessment Tools

*Text reading tasks* (Prove MT-3-Clinica [45]): These tasks consisted of informative texts for students to read aloud. Text length ranges from 345 to 1287 syllables (168 to 545 words) and increases with grade level. Different texts were used for each class. The texts used for this task were not used for the text reading comprehension task. Performance was calculated as mean reading speed (syllables per second). As reported in the manual, reading speed test–retest indices ranged from 0.85 to 0.96. 

*Text comprehension tasks* (Prove MT-3-Clinica [45]): The tasks consisted of one narrative and one informative text pair. In the pre-test evaluation, each child was randomly presented with an informative or narrative text, whereas those not presented were used in the post-test assessment. Texts ranged from 226 to 455 words in length and increased with grade level. The children were asked to read the text silently at their own pace, answer 12 multiple-choice questions, and choose 1 out of 4 possible answers. There was no time limit and children could reread the text at any time. The total score is calculated from the total number of correct answers. As stated in the manual, the alpha factor ranged from 0.61 to 0.83.

*Word and non-word reading tasks* (DDE-2 [46]): These tasks were taken from a standardized battery for the assessment of developmental dyslexia and dysgraphia in elementary school. Lists of words and non-words were given to children and they were asked to read them aloud. Non-words consist of strings of letters that appear similar to words but have no meaning. The time taken to read the list and the number of mistakes made (which were not corrected) were recorded. Reading performance was measured using average reading speed. The DDE-2 task consisted of sequences of 84 words and 48 non-words containing 2–4 syllables, for a total of 281 and 127 syllables, respectively. Test–retest reliability for word and non-word reading speed was around 0.80 in many studies (e.g., Cornoldi et al. [47]).

*Text dictation* (BVSCO-2, Batteria per la valutazione della scrittura e della competenza ortografica (Battery for the Assessment of Writing Skills of Children from 7 to 13 years of age) [48]): These tasks consisted of a text that was read aloud from the professional and the children were asked to write it down. For each of the classes, a passage was chosen that varied in content, syntactic complexity, and, above all, frequency of use of words. A different text was used for each grade.

*Word and non-word dictation* (DDE-2 [46]): In these tasks, lists of words and non-words are read to the children, one word at a time. Children have to write down every word or non-word they hear. The task consists of 48 words and 24 non-words containing between two and four syllables.

### 2.4. Remote Assessment Tool

TeleFE [44], for the remote evaluation of executive functions: It is a tool that allows the evaluation, in person or remotely, of working memory updating, cognitive flexibility, inhibition, and planning. Inside there are four tests and two questionnaires, three of the tests were proposed to all children in the present research and are described below.

*Flanker Test* [49]: An activity to measure interference control and cognitive flexibility, i.e., the child’s ability not to be distracted by irrelevant information and to shift his/her attention to follow two simultaneous rules. In this task, strings of five aligned arrows appear on the screen and the child is required to indicate the direction of the target arrow and to ignore the others. The target arrows could indicate the same direction as the others (congruent condition) or the opposite direction (incongruent condition). The task consists of three blocks. The 1st and 2nd blocks are single rule tasks, asking the child to indicate, in the 1st block, the direction of the arrow in the center, and in the 2nd block the direction of the external arrows. Finally in the 3rd block, called mixed rule, both above-mentioned conditions are presented. The measures considered were the accuracy of the incongruent condition of the single rule task and the incongruent condition of the mixed rule task.

*Go-No/Go test* [50]: An activity to measure inhibition, understood as the skill to inhibit an automatic response. The child is asked to press a key, as fast as he can, only when he sees a figure of a certain type and not to press when he sees a different one. A sequence of geometrical shapes (yellow or blue triangles or circles) is displayed on the screen and children are advised to respond to target stimuli and not to non-target stimuli. The task consists of 4 blocks containing 50 items (35 Goes and 15 NoGos). In the first block, Go stimuli are yellow shapes (regardless of geometry) and NoGo stimuli are blue shapes. The second block reverses the pattern. In the third block, Go stimuli are circles (regardless of color) and NoGo stimuli are triangles. The fourth block inverts the pattern. The measure considered is responses to the Go stimuli (Go CR);

*N-Back test* [51]: An activity to measure working memory, i.e., the ability to keep and operate on material temporarily held in memory. The child sees a series of stimuli in the center of the screen and presses the spacebar when the stimulus matches one of the previous stimuli. The task consists of three different conditions: color, shape, and letter. Each condition has two blocks, 1-back and 2-back, respectively, for a total of 6 different blocks. The measures are the number of correct answers for 1-back block and the number of correct answers for 2-back blocks.

### 2.5. Intervention Tools

For the intervention, it was decided to use the Anastasis RIDInet platform, an online rehabilitation platform designed to facilitate and increase the effectiveness of the rehabilitation intervention on SLD. At the basis of this tool, there is a scientific committee whose members include SLD experts. The tool is aimed at trained clinicians such as psychologists, speech therapists, or neuropsychiatrists.

The available activities are developed according to recognized rehabilitation models and are controlled by the specialist who assigns his patients’ rehabilitation activities to be carried out at home, to complement the outpatient intervention. The apps used for the present study were:

*Reading Trainer 2* [52] is a software aimed at treating reading disorders. The options it is equipped with make it easier to read single graphemes, syllables, morphemes, and words, both in isolation and within sentences and passages, favoring both visual-perceptive and attentive processes, and phonological ones, for making the association between graphemes and phonemes more accurate and faster. This exercise allows intervention on the skill that directly stimulates the competence compromised. The exercise is intensive (the required time is about 20 min per day, at least 3 times per week for a total treatment duration of about 3 months) and self-adjusting, and provides remote clinician monitoring; the clinician, based on the results obtained from the performance of the child, can modify and customize some parameters, such as the scan speed, the reading unit (syllabic or sub-lexical), the font, and also the complexity of the text and words used in the text [53].

*Run the Ran* [37] is aimed at enhancing the rapid naming processes considered important for the acquisition of reading both in typical development and in developmental dyslexia. It is aimed at speeding up the sequential retrieval of the phonological labels corresponding to different visual stimuli. The app aims to enhance the prerequisites of reading by requiring the timed and progressively faster naming of color matrices or black and white figures; the stimuli are organized according to the type of stimulus (the stimuli are organized in libraries according to the length, complexity, and semantic categories of the corresponding words). This telerehabilitation program requires training for about 15 min at least 3 times a week.

*Cloze 2* [54] promotes the ability to understand the written text and the recovery of lexical and semantic inference processes. The child must work on a wordlessly written text and fill in the blanks by choosing the correct alternative from the alternatives automatically suggested by the app to match the text. The app is aimed at children and young people who have a weakness in understanding the written text, both in the presence of a condition of text comprehension disorder and in the presence of poor skills in lexical and semantic inferential processes.

*MemoRAN* [55] works on basic executive functions starting from rapid visual naming exercises: the task is to name multiple visual stimuli (images) presented in matrices faster and faster, leveraging a visual–verbal integration process for rapid lexical access. This is associated with secondary tasks involving various executive functions: inhibition, working memory, and cognitive flexibility.

### 2.6. Statistical Analysis

The descriptive and inferential statistics were conducted using the Statistical Package for Social Sciences 2022, version 28.0.1.0. Analysis of the normality of the distribution (skewness cut-off = 2; kurtosis cut-off = 3) was carried out on all measures. Due to the non-normality of some variables (accuracy in word and non-word reading), the logarithmic transformation was conducted and Welch correction for the analysis of variance was used.

In the first part, paired *t*-tests were conducted in the training and waiting list groups separately, then a multivariate analysis of variance of Group (training vs. waiting list) × Session (pre vs. post) was conducted, to compare the group of children who underwent the telerehabilitation intervention with the children on the waiting list. Subsequently, a subgroup of children who used the RT-2 combined with the Run the Ran app was compared with children in the waiting list group. A paired *t*-test and multivariate analysis of variance Group (RT-2 + Run the Ran vs. waiting list) × Session (pre vs. post) was conducted.

In the second part, the variables of the number of sessions of the apps proposed and of the executive functioning performances of the children were considered to analyze if, and how, these aspects can correlate with the changes in learning abilities. A one-way analysis of variance was run to compare the deltas (results at post-test minus results at pre-test) of executive functions between the training and the waiting list group. A correlation between the number of sessions in the two apps separately and the deltas of learning abilities was conducted. Furthermore, considering the results at TeleFE, a correlation between deltas at the EF performances and deltas at learning performances was conducted.

## 3. Results

### 3.1. Comparison between Training and Waiting List Group

At the pre-test, the two groups were homogeneous except for errors in text reading, in which the training group initially had a greater number of errors than the waiting list group (M = 14.85 (7.73) vs. M = 10.15 (6.14)).

Because of the small differences identified in the initial assessment between the training and waiting list groups of children with SLD (Table 1), it was decided to study the changes in the performance of the two groups separately (paired sample Student’s *t*-test with Bonferroni correction). Furthermore, a 2 × 2 multivariate analysis of variance with Group (training vs. waiting list) as the between-group variable, and Session (pre-test vs. post-test) as a within-group factor, was run on the performance obtained on each measure (Table 1).

Considering text reading fluency, the main effects of Session (F_1.71_ = 29.09, *p* < 0.001, *η*^2^*_p_* = 0.291) were significant, and the performance at the pre-test was lower than that at the post-test. The effect of Group (F_1.71_ < 1) was not significant, as well as the Session × Group interaction (F_1.71_ < 1). In text reading accuracy, only the effect of the Group was significant (F_1.58_ = 6.37, *p* = 0.014, *η*^2^*_p_* = 0.10), with the training group having a high number of errors. The effect of the Session and the interaction were not significant (F_1.58_ < 1).

Word reading fluency showed the main effect of Session (F_1.65_ = 69.92, *p* < 0.001, *η*^2^*_p_* = 0.517). The effect of Group (F_1.65_ < 1) was not significant. The Session × Group interaction was significant (F_1.65_ = 5.58, *p* = 0.021, *η*^2^*_p_* = 0.079), with children with SLD in the training group showing better performance in the post-test compared to children with SLD in the waiting list group. Non-word reading fluency, and word and non-word reading accuracy showed the main effects of Session (respectively F_1.66_ = 21.21, *p* < 0.001, *η*^2^*_p_* = 0.243; F_1.60_ = 15.81, *p* < 0.001, *η*^2^*_p_* = 0.209; F_1.63_ = 16.79, *p* < 0.001, *η*^2^*_p_* = 0.210) but no Group or interactions effects (F < 1) were significant.

Text dictation accuracy showed the main effect of Session (F_1.69_ = 14.44, *p* < 0.001, *η*^2^*_p_* = 0.173). The effect of Group (F_1.65_ < 1) was not significant. The Session × Group interaction was significant (F_1.69_ = 4.48, *p* = 0.038, *η*^2^*_p_* = 0.061) with children with SLD in the training group showing better performance at the post-test compared to children with SLD in the waiting list group.

Word and non-word dictation showed the main effect of Session (respectively F_1.69_ = 10.16, *p* = 0.002, *η*^2^*_p_* = 0.135; F_1.67_ = 16.99, *p* < 0.001, *η*^2^*_p_* = 0.202) but no Group or interactions effects (F < 1) were significant. Finally, for reading comprehension, no effects resulted in being significant.

Analyzing the effect size, the results showed that the trained group significantly improved in text reading fluency more than the waiting list group (*d* = 0.31 vs. *d* = 0.24), which showed the normal trend expected for SLD children [34]. Similarly, considering word and non-word reading fluency (*d* = 0.49 vs. *d* = 0.31 and *d* = 0.34 vs. *d* = 0.25, respectively), the training group showed a greater improvement in reading speed compared to the waiting list group. Considering accuracy, children in the training group showed a greater effect size [56] compared to children in the waiting list group in text reading accuracy (*d* = 0.37 vs. *d* = 0.07), word reading accuracy (*d* = 0.33 vs. *d* = 0.23), and no-word reading accuracy (*d* = 0.36 vs. *d* = 0.35). Finally, considering text dictation, a greater effect size emerged in the case of the training group (*d* = 0.40 vs. *d* = 0.23), except for word dictation (*d* = 0.39 vs. *d* = 0.40) and non-word dictation (*d* = 0.31 vs. *d* = 0.55) in which the waiting list group showed greater effect sizes.

### 3.2. Comparison between Training Reading and RAN and Waiting List Group

Within the training group, a subgroup of children used the Reading Trainer 2 and Run the Ran apps. This subgroup was compared to the waiting list group. The two groups were comparable in terms of age (χ^2^ = 0.18, *p* > 0.05) and gender (χ^2^ = 0.21, *p* > 0.05). At the pre-test variables, the two groups were homogeneous, except for the variables in text reading fluency and accuracy and word reading fluency, in which children attending the training had worse performances compared to children in the waiting list group.

A paired sample Student’s *t*-test with Bonferroni correction and a 2 × 2 multivariate analysis of variance with Group (RT2 + Run the Ran vs. waiting list) as the between-group variable, and Session (pre-test vs. post-test) as a within-group factor, was run on the performance obtained on each measure (Table 2). Variables that differed at the pre-test were used as covariates in the multivariate analysis.

Considering text reading fluency, the main effects of Session (F_1.43_ = 35.44, *p* < 0.001, *η*^2^*_p_* = 0.452) were significant, and the performance at the pre-test was lower than that at the post-test. The effect of Group (F_1.43_ < 1) was not significant, as well as the Session × Group interaction (F_1.43_ < 1). In text reading accuracy, the effect of the Session was significant (F_1.37_ = 6.26, *p* = 0.017, *η*^2^*_p_* = 0.145), as well as the effect of the Group (F_1.37_ = 13.03, *p* = 0.001) with the training group having a high number of errors, and the interaction was not significant (F_1.37_ =2.57, *p* = 0.118).

Word reading fluency showed the main effect of Session (F_1.39_ = 82.20, *p* < 0.001, *η*^2^*_p_* = 0.678). The effect of the Group was not significant (F_1.39_ < 1) but the Session × Group interaction (F_1.39_ = 12.24, *p* = 0.001) was, with children with SLD attending the training with RT-2 and Run the Ran showing better performance at the post-test compared to children with SLD in the waiting list group.

Non-word reading fluency, and word and non-word reading accuracy showed the main effect of Session (respectively, F_1.40_ = 13.33, *p =* 0.001, *η*^2^*_p_* = 0.250; F_1.37_ = 8.33, *p =* 0.006, *η*^2^*_p_* = 0.184; F_1.38_ = 17.74, *p* < 0.001, *η*^2^*_p_* = 0.318), but no Group or interaction effects (F < 1) were significant.

Text dictation accuracy showed the main effect of Session (F_1.43_ = 18.32, *p* < 0.001, *η*^2^*_p_* = 0.299). The effect of Group (F_1.43_ < 1) was not significant. The Session × Group interaction was significant (F_1.43_ = 6.95, *p* = 0.012, *η*^2^*_p_* = 0.139) with children with SLD attending the training with RT-2 and Run the Ran showing better performance at the post-test compared to children with SLD in the waiting list group.

Word and non-word dictation showed the main effect of Session (respectively F_1.43_ = 4.69, *p* = 0.036, *η*^2^*_p_* = 0.103; F_1.42_ = 9.77, *p* = 0.003, *η*^2^*_p_* = 0.185), but no Group or interaction effects (F < 1) were significant. Finally, for reading comprehension, no effects resulted in being significant.

Analyzing the effect size, the results showed that the trained group significantly improved in text reading fluency more than the waiting list group (*d* = 0.60 vs. *d* = 0.24), which showed the normal trend expected for SLD children [34]. Similarly, considering word and non-word reading fluency (*d* = 0.67 vs. *d* = 0.31 and *d* = 0.40 vs. *d* = 0.25, respectively), the training group showed a greater improvement in reading speed compared to the waiting list group. Considering accuracy, children in the training group showed a greater effect size [56] compared to children in the waiting list group in text reading accuracy (*d* = 0.58 vs. *d* = 0.07), word reading accuracy (*d* = 0.49 vs. *d* = 0.23), and non-word reading accuracy (*d* = 0.40 vs. *d* = 0.35). Finally, considering text dictation, a greater effect size emerged in the case of the training group (*d* = 0.62 vs. *d* = 0.23), except for word dictation (*d* = 0.21 vs. *d* = 0.40) and non-word dictation (*d* = 0.18 vs. *d* = 0.55) where the waiting list group showed greater effect sizes.

### 3.3. Number of Sessions and Differences in Learning

Considering the number of sessions for each of the two apps the children used during the training, the correlations with the differences between the post- and pre-tests of each learning measure were considered. Children of the training group showed a mean number of sessions for the first app of M = 33.90 (15.42) and the second app of M = 28.06 (14.77). As shown in Table 3 the number of sessions on the second app, concerning processes (executive functions), had a negative significant correlation with Δ of text reading accuracy (indicating errors).

### 3.4. Executive Functions and Differences in Learning

Considering the remote assessment of executive functions, Table 4 shows the descriptive statistics of the two groups at the pre- and post-test and the results of the 2 × 2 analysis of variance with Group (training vs. waiting list) as the between-group variable, and Session (pre-test vs. post-test) as a within-group factor. The results of the ANOVA showed a significant interaction in Δ of the correct responses at the incongruent condition in the single rule and in the mixed rule, with children with SLD attending the training performing better at post-test compared to children on the waiting list.

Considering the correlations between Δ of more significant measures of TeleFE [57] and Δ of learning tasks, a significant negative correlation emerged between Δ word reading accuracy and Δ of correct responses on the Flanker mixed rule incongruent stimuli condition (r = −0.51, *p* < 0.05).

## 4. Discussion

SLD are multidimensionally challenging conditions that affect academic abilities and cognition, and thus it is important to address both the learning skills and the cognitive functions underlying the conditions through effective training. It is recognized that an absence of early and efficacious intervention can affect the well-being of individuals with SLD and their functioning and social life [58]. For SLD, the main areas of difficulty are usually decoding words. Reading difficulties can cause problems in other academic areas such as math, science, and social studies. Additionally, many people with SLD can have difficulty organizing, managing, and setting reasonable goals for their time. These can be carried out remotely, which is why telerehabilitation is suitable. Over the past decade, the use of telerehabilitation managed by certified clinicians has helped ameliorate difficulties (related to literacy) and diminish training costs [59]. Furthermore, recent studies have shown the efficacy of tele-intervention software for children with SLD [10,35,36,53], showing that telerehabilitation, which aims to integrate into complex activities literacy and a variety of cognitive, verbal, visual, and attentional processes, is promising.

The purpose of this study was to evaluate the effectiveness of telerehabilitation training in children with SLD, between the third and fifth year of primary school, through a platform that allows the use of two apps at a time. The first app is specifically designed for training an academic skill (reading, writing, comprehension, or calculation), and the second app aims to train the processes underlying learning (rapid automated naming, working memory, scanning left–right). The child and his/her family were introduced to the platform by the clinician who then remotely followed the child’s exercises and met him/her in attendance every three weeks to monitor the intervention. The activities were conducted at home by the child, accompanied by an adult, for about 5 sessions per week for 15/20 min for a total period of 3 months. The results showed that children who carried out the training improved their fluency in reading and their performance (less errors) in text dictation, confirming previous studies [31], with greater effect sizes compared to children in the waiting list group that showed a normal trend expected in SLD [34]. Similar results emerged for children who trained on the Reading Trainer 2 app in conjunction with the Run the Ran app. In both training groups, fewer changes emerged in word and non-word dictation. The improvement in text dictation but not in word and non-word dictation can be explained by the greater context support in the case of the former than in the latter.

The correlations between the number of training sessions and the change between pre- and post-test results in learning skills show a correlation between the number of sessions on the “process” app (rapid automatized naming or working memory) and the accuracy of text reading. In line with other current work, there are no other significant correlations between the number of sessions and the improvement in learning [31]. Concerning executive functions, comparing the deltas performance of children who were a part of the training and waiting list groups, the results revealed a change in the Flanker test, in both the single rule and mixed rule modes. The test evaluates interference control and cognitive flexibility, and the results showed a better performance of the training group at post-test compared to the waiting list group. Finally, in the correlations between the deltas of the performance in executive functions and the deltas in the performance of learning abilities, the parameter of text reading accuracy emerged as correlating with the control of the interference and the cognitive flexibility.

Thus, considering the literature present on the telerehabilitation of SLD, it emerges that the effect of telerehabilitation on reading ability shows improvements above all in reading speed [10,33,35], and in some cases in reading accuracy [10,35]. Similarly, it emerges that the effect of telerehabilitation on executive functions shows improvements in these abilities [26,38,60]. Maggio and colleagues [60] found that the use of telerehabilitation that combined training for specific learning deficits with executive functions, in an adolescent sample with SLD, was associated with an increase in cognitive skills. In the present study, both aspects emerge, the combination of an app on the learning task and an app on general processes (RAN, working memory) shows improvement in both aspects. Furthermore, in the present work, there is an improvement in the accuracy of text dictation. This evidence would seem to support two hypotheses: the hypothesis that the enhancement of the decoding process could also allow the recovery of writing performance [61] and the hypothesis that work on the executive functions could facilitate the monitoring and control of interference, acting on the improvement in accuracy [62].

The clinical implications concern, therefore, several aspects: the importance of an assessment of executive functions as well as learning in children with SLD, giving space in telerehabilitation to work on these aspects as they can influence learning, considering telerehabilitation as a valid tool to effectively work by combining aspects related to learning with aspects related to processes (more difficult with activities in person). From a clinical and practical point of view, an important implication is that telerehabilitation has allowed the public services involved in the study to remove the waiting lists and to guarantee a timely take-over, fundamental in the case of SLD [21].

The implications of this study in terms of research concern the importance of analyzing the association between different apps on learning (reading, writing, comprehension, calculation) with different apps on executive functions. Furthermore, it is important to consider whether and which pairing could be more effective based on the child’s profile. Future frontiers in research in this area concern the possibility of also working with telerehabilitation in the school context [63] to make the intervention for children with SLD even more compact and integrated.

This line of research, which sees telerehabilitation at the forefront, supported by face-to-face monitoring, appears to be the future direction in the treatment of SLD due to its practicality of use, and the unique possibility of personalization and self-adaptive activities, which are fundamental for allowing the progressive change that each individual with SLD shows differently from the other [31].

### Limitations and Future Directions

The main limitation of this study generally concerns efficacy in clinical research studies and concerns the creation of the experimental group; in fact, testing small samples is often inevitable in studies regarding neurodevelopmental disorders, and often these are identified by different procedures (e.g., teacher reporting, clinical procedures, etc.). The consequence of this aspect is that randomized controlled studies are infrequent and to this is added the difficulty of collecting data over time (follow-up). It follows that, as in the present study, the effects in this field are small [64]. Therefore, another limitation of this study is the small, heterogeneous, in terms of different learning disabilities, sample with the consequence that the results cannot be extended to the entire population of patients with SLD. Due to the not so large number of children, children with SLD were considered as a unique group. In future studies, the possibility of expanding the sample may offer the possibility of separately analyzing children with and without comorbidities in learning disabilities. In future studies, it will also be important to evaluate the degree of acceptance and satisfaction of the professionals involved in this type of intervention [65] to ensure that clinicians believe in the treatment they propose and that a serene working climate is maintained. Finally, a key aspect that needs to be addressed concerns the lack of comprehensive research that provides evidence to support decision makers and policy makers in adopting telerehabilitation techniques in clinical settings as well as the general lack of standardization in the terms used [66]. 

## 5. Conclusions

In conclusion, this study suggests that telerehabilitation training may be useful for improving learning abilities and executive function skills in children with SLD. Telerehabilitation results in being useful and effective in implementing reading and writing skills, with greater improvements than expected by the normal development trend of children with SLD. The combination of an app on the academic task and an app on the processes seems to facilitate, in addition to the reading speed, the accuracy in both reading and text dictation. In addition, this innovative system can facilitate children’s home care to meet their individual needs and ensure continuity of care and timely rehabilitation interventions, especially during lockdowns.

## Figures and Tables

**Table 1 brainsci-13-00479-t001:** Performance at pre- and post-test of children with SLD in the training and waiting list conditions and MANOVA results for the Group * Session interaction.

	Training			Waiting List			
	PRE M (SD)	POST M (SD)	Paired Sample *t*-test	PRE M (SD)	POST M (SD)	Paired Sample *t*-test	MANOVA Group * Session
Text reading fluency (syll/sec)	1.84 (0.76)	2.08 (0.79)	*t*(47) = −5.42, *p* < 0.001 *	1.91 (0.73)	2.09 (0.75)	*t*(24) = −2.19, *p* = 0.039	*F*(1.71) = 0.61, *p* = 0.437 *η*^2^*_p_* = 0.009
Text accuracy (err)	14.85 (7.73)	12.23 (6.58)	*t*(39) = 2.43, *p =* 0.020	9.84 (5.41)	9.38 (6.48)	*t*(20) = 0.36, *p* = 0.722	*F*(1.58) = 0.35, *p* = 0.556 *η*^2^*_p_* = 0.006
Word reading fluency (syll/sec)	1.65 (0.76)	2.03 (0.80)	*t*(43) = −7.26, *p* < 0.001 *	1.74 (0.69)	1.96 (.72)	*t*(22) = −4.44, *p* < 0.001 *	*F*(1.65) = 5.58, *p* = 0.021 * *η*^2^*_p_* = 0.079
Word accuracy (err)	9.27 (6.79)	6.99 (7.17)	Z = −3.28, *p =* 0.001 *	10.0 (6.81)	8.54 (5.76)	*t*(23) = −3.31, *p =* 0.003 *	*F*(1.60) = 1.49, *p* = 0.227 *η*^2^*_p_* = 0.024
Non-word reading fluency (syll/sec)	1.12 (0.41)	1.26 (0.41)	*t*(43) = −3.90, *p* < 0.001 *	1.22 (0.53)	1.36 (.57)	*t*(23) = −1.90, *p =* 0.070	*F*(1.66) = 0.07, *p* = 0.794 *η*^2^*_p_* = 0.001
Non-word accuracy (err)	10.67 (6.02)	8.41 (6.49)	Z = −2.97, *p =* 0.003 *	10.21 (6.52)	7.79 (7.24)	Z = −2.31, *p =* 0.020	*F*(1.63) = 0.08, *p* = 0.776 *η*^2^*_p_* = 0.001
Text dictation (err)	25.06 (15.53)	19.13 (14.35)	*t*(46) = 4.60, *p* < 0.001 *	22.52 (11.42)	20.04 (9.67)	*t*(24) = 1.71, *p =* 0.099	*F*(1.69) = 4.48, *p* = 0.038 * *η*^2^*_p_* = 0.061
Word dictation (err)	9.87 (7.12)	7.23 (6.32)	*t*(44) = 2.99, *p =* 0.005 *	8.76 (5.14)	6.92 (3.99)	*t*(24) = 2.21, *p =* 0.037	*F*(1.69) = 4.48, *p* = 0.768 *η*^2^*_p_* = 0.001
Non-word dictation (err)	5.80 (3.17)	4.84 (2.99)	*t*(44) = 2.56, *p =* 0.014	5.88 (2.11)	4.44 (2.98)	*t*(24) = 3.42, *p =* 0.002 *	*F*(1.67) = 2.26, *p* = 0.137 *η*^2^*_p_* = 0.033
Accuracy text comprehension	5.02 (2.27)	5.41 (2.54)	*t*(43) = −0.89, *p =* 0.378	5.40 (2.48)	4.96 (2.72)	*t*(24) = 0.71, *p=* 0.482	*F*(1.66) = 0.45, *p* = 0.506 *η*^2^*_p_* = 0.007

* Significant (alpha= 0.05) in paired sample *t*-test after Bonferroni’s correction (*p* = 0.005).

**Table 2 brainsci-13-00479-t002:** Performance at pre- and post-test of children with SLD attending RT-2 + Run the Ran apps and waiting list group and results of the MANOVA for the Group * Session interaction.

	RT-2 + Run the Ran			Waiting List			
	PRE M (SD)	POST M (SD)	Paired Sample *t*-test	PRE M (SD)	POST M (SD)	Paired Sample *t*-test	MANOVA Group * Session
Text reading fluency (syll/sec)	1.48 (0.55)	1.85 (0.67)	*t*(19) = −5.89 *p* < 0.001 *	1.91 (0.73)	2.09 (0.75)	*t*(24) = −2.19 *p* = 0.039	*F*(1.43) = 1.63 *p* = 0.209 *η*^2^*_p_* = 0.104
Text reading accuracy (err)	19.26 (7.30)	14.68 (8.34)	*t*(18) = 2.51 *p* = 0.022	9.84 (5.41)	9.38 (6.48)	*t*(20) = 0.36 *p* = 0.722	*F*(1.37) = 2.57 *p* = 0.118 *η*^2^*_p_* = 0.065
Word reading fluency (syll/sec)	1.29 (0.55)	1.70 (0.66)	*t*(17) = −6.68 *p* < 0.001 *	1.74 (0.69)	1.96 (0.72)	*t*(22) = −4.44 *p* < 0.001 *	*F*(1.39) = 12.24 *p =* 0.001 * *η*^2^*_p_* = 0.239
Word reading accuracy (err)	12.05 (6.87)	8.97 (5.69)	*t*(17) = 2.44 *p* = 0.026	10.00 (6.81)	7.54 (5.76)	*t*(23) = 3.31 *p* = 0.003 *	*F*(1.37) = 0.38 *p* = 0.541 *η*^2^*_p_* = 0.010
Non-word reading fluency (syll/sec)	0.98 (0.35)	1.12 (0.35)	*t*(17) = −2.13 *p* = 0.048	1.22 (0.53)	1.36 (0.57)	*t*(23) = −1.90 *p* = 0.070	*F*(1.40) = 0.28 *p* = 0.598 *η*^2^*_p_* = 0.007
Non-word reading accuracy (err)	12.67 (6.07)	9.22 (7.01)	*t*(17) = 2.57 *p* = 0.020	10.21 (6.52)	7.79 (7.24)	Z = −2.31 *P* = 0.020	*F*(1.38) = 0.75 *p* = 0.391 *η*^2^*_p_* = 0.019
Text dictation (errors)	34.20 (15.70)	24.65 (15.16)	*t*(19) = 5.77 *p* < 0.001 *	22.52 (11.42)	20.04 (9.67)	*t*(24) = 1.71 *p* = 0.099	*F*(1.43) = 6.94 *p* = 0.012 * *η*^2^*_p_* = 0.139
Word dictation (errors)	10.85 (6.81)	9.50 (6.21)	*t*(19) = 1.87 *p* = 0.077	8.76 (5.14)	6.92 (3.99)	*t*(24) = 2.21 *p* = 0.037	*F*(1.41) = 0.36 *p* = 0.550 *η*^2^*_p_* = 0.009
Non-word dictation (errors)	5.85 (3.13)	5.30 (3.13)	*t*(19) = 1.09 *p* = 0.290	5.88 (2.11)	4.44 (2.98)	*t*(24) = 3.42 *p =* 0.002 *	*F*(1.43) = 2.43 *p* = 0.126 *η*^2^*_p_* = 0.054
Accuracy text comprehension	4.55 (2.41)	4.45 (2.32)	*t*(19) = 0.20 *p* = 0.841	5.40 (2.48)	4.96 (2.72)	*t*(24) = 0.71 *p* = 0.482	*F*(1.42) = 0.04 *p* = 0.849 *η*^2^*_p_* = 0.001

* Significant (alpha= 0.05) in paired sample *t*-test after Bonferroni’s correction (*p* = 0.005).

**Table 3 brainsci-13-00479-t003:** Correlations between the number of sessions of the two apps and Δ of learning tests.

	N. Sessions App 1	N. Sessions App 2
Δ of text reading fluency (syll/sec)	0.227	0.036
Δ of text reading accuracy (err)	−0.177	−0.366 *
Δ of word reading fluency (syll/sec)	0.148	0.074
Δ of word reading accuracy (err)	−0.158	−0.060
Δ of non-word reading fluency (syll/sec)	0.006	0.101
Δ of non-word reading accuracy (err)	−0.190	−0.076
Δ of text dictation (err)	0.142	−0.055
Δ of word dictation (err)	−0.017	−0.008
Δ of non-word dictation (err)	−0.023	−0.063
Δ of text comprehension (correct responses)	−0.185	−0.021

* The correlation is significant at the value of 0.05.

**Table 4 brainsci-13-00479-t004:** Executive functions’ performance at pre- and post-test of children with SLD in the training and waiting list conditions and results of ANOVA comparing Δ.

	Training			Waiting List			
	PRE M (SD)	POST M (SD)	Delta M (SD)	PRE M (SD)	POST M (SD)	Delta M (SD)	Anova Group * Session
NoGo CR	11.22 (2.63)	11.14 (2.38)	−0.08 (2.56)	10.00 (2.49)	10.52 (1.76)	0.98 (2.29)	*F*(1.33) = 1.18 *p* = 0.285 *η*^2^*_p_* = 0.035
Flanker single Inc.	61.03 (26.15)	80.45 (24.91)	16.52 (26.91)	79.08 (13.07)	73.08 (17.52)	−6.00 (19.45)	F(1.32) = 6.85 *p* = 0.013 * *η*^2^*_p_* = 0.176
Flanker mixed Inc.	45.72 (22.78)	60.10 (19.07)	12.15 (20.15)	55.61 (17.32)	51.00 (20.04)	−4.61 (15.47)	*F*(1.31) = 6.48 *p* = 0.016 * *η*^2^*_p_* = 0.173
1-back	11.50 (5.63)	10.59 (5.91)	−0.65 (4.34)	11.47 (2.96)	11.19 (4.74)	−0.67 (4.61)	*F*(1.28) = 0.00 *p* = 0.993 *η*^2^*_p_* = 0.001
2-back	7.80 (3.76)	7.49 (5.24)	0.34 (2.98)	6.93 (3.00)	5.34 (4.33)	0.00 (3.34)	*F*(1.27) = 0.06 *p* = 0.886 *η*^2^*_p_* = 0.002

* The correlation is significant at the value of 0.05.

## Data Availability

The data presented in this study are available on request from the corresponding author.

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
