# Peer review of "Telerehabilitation Pathways in Specific Learning Disorders: Improving Reading and Writing"

_brainsci, 2023, doi:10.3390/brainsci13030479_

Round 1
Reviewer 1 Report
Dear Authors, I sincerely congratulate you on your contribution to the study. The article presents high cognitive value and is in keeping with the nature of the journal. Actually, I do not make any comments in the methodological and research layers. However, I would ask the Authors to expand the discussion, because in its current form it does not provide a chance to acquaint the reader with other developments in the field of the paper. Also, please rewrite the conclusions, as the current ones are very broad, perhaps a few synthetic points would suffice?
I also hope that the Authors note the limitations of their study, such as the small sample size. Please indicate at the end of the discussion any difficulties or limitations observed during your research. Such practice will allow correct adoption of the methodology in future studies.
Greetings!
Author Response
We thank the reviewer very much for highlighting the positive and interesting aspects of the work
I would ask the Authors to expand the discussion, because in its current form it does not provide a chance to acquaint the reader with other developments in the field of the paper. Also, please rewrite the conclusions, as the current ones are very broad, perhaps a few synthetic points would suffice?
Thank you for your suggestion. We expand the discussion section providing a more extensive view of the literature on the topic. We also rewrite a more synthetic conclusion with a greater focus on the main results of the study.
I also hope that the Authors note the limitations of their study, such as the small sample size. Please indicate at the end of the discussion any difficulties or limitations observed during your research. Such practice will allow correct adoption of the methodology in future studies.
A section about ‘Limit and future directions’ has been added, and the small sample size and other issues have been considered.
Reviewer 2 Report
15 February 2023
Manuscript ID: brainsci-2243713
Type: Review
Title: ‘Effectiveness of telerehabilitation pathways in Specific Learning Disorders’ by Capodieci A et al., submitted to Brain Sciences
Dear Authors,
Telerehabilitation applies information and communication technologies to provide rehabilitation services to people who reside remotely. In the present research article entitled ‘Effectiveness of telerehabilitation pathways in Specific Learning Disorders’, Capodieci and colleagues investigated the components of learning capacities which benefit from telerehabilitation in children with specific learning disorders.
The main strength of this manuscript is that it addresses an interesting and timely question, providing a captivating interpretation addressing the improvement of reading fluency and accuracy and text dictation in the training group, compared to those on the waiting list.
In general, I think the idea of this article is really interesting and the authors’ fascinating observations on this timely topic may be of interest to the readership of Brain Sciences. However, some comments, as well as some crucial evidence that should be included to support the author’s argumentation, needed to be addressed to improve the quality, the adequacy, and the readability of the manuscript prior to the publication. My overall opinion is to publish this paper after the authors have carefully considered my suggestions below, in particular reshaping parts of the ‘Introduction’ and ‘Discussion’ sections by adding more evidence.
Please consider the following comments:
1. Title: Please present the self-explanatory title stating the most significant results of this study.
2. Abstract: Using up to 200 words, please proportionally present the background, the objectives, the methods, the results, and the conclusion. The background should include the general background (one to two sentences), the specific background (two to three sentences), and current issue addressed to this study (one sentence). Especially, the passage “The analysis of some of the variables that may or may not affect 11 the effectiveness of the route is being studied.” reads obscure. I would like the authors to clarify it in the specific background and in the current issue addressed to this study. The result should close with one to two sentences which put the result into a more general context. The conclusion should include one sentence describing the main message using such words like “Here we show”, the potential and the advance this article has provided in the field, and finally a broader perspective (two to three sentences) readily comprehensible to a scientist in any discipline. So, I would like the authors to elaborate the abstract accordingly.
3. Keywords: According to the Journal’s guidelines, please list ten pertinent keywords, which I suggest being specific to the article, yet reasonably common within the subject discipline. Also, please as many as possible in the title and in the first two sentences of the abstract
4. A graphical abstract that will visually summarize the main findings of the manuscript is highly recommended.
5. Introduction: I believe that this section would benefit from reorganizing this section by focusing the background (general, specific, and the issue addressed to this study) and providing sufficient information on the main constructs of this topic, leading to the objectives. Those constructs should be acknowledgeable to any reader in any discipline and should make this section persuasive enough to put forward the main purpose of current research the authors conduct and the specific purpose the authors have intended by this paper. Those main structures should be organized in a logical and cohesive manner.
I think that more organized and detailed information on specific learning disorders would provide a suitable background here. Thus, I suggest the authors to make their effort to provide a brief overview of the pertinent literature that offer a perspective on definition, causes and symptoms of specific learning disorder, because as it stands, this information is not highlighted in the text. The background should be presented in the following order: schizophrenia in general including brief descriptions of epidemiology, pathogenesis, symptoms, current treatment, and challenge in treatment, comorbidity in specific learning disorders such as attention deficit hyperactivity disorder, conduct disorder, anxiety, and depression. In this regard, I would suggest providing a general overview of pathogenesis and biochemical hallmarks of specific learning disorders, for example ‘mitochondrial impairment as a common motif in neuropsychiatric presentation’ and ‘modelling the neurodevelopmental pathogenesis in neuropsychiatric disorders’. Moreover, I would recommend adding more information on neural substrates of specific learning disorders, specifically on frontal lobe dysfunction, and on related effects on patients’ memory and learning impairments, as ‘Functional interplay between central and autonomic nervous systems’ in this disorder: this information may provide a better understanding of prefrontal cortex’s key role and how its disrupted function may contribute to irregular behavioral responses (doi:10.17219/acem/139572) and therefore to the development of many cognitive dysfunctions (i.e., impaired memory, attention, working memory, problem solving, processing speed, and social cognition) that are common in schizophrenia (https://doi.org/10.3389/fnbeh.2022.998714).
6. Methods: I would like the authors to open this section with an introductory paragraph to help a reader understand the flow of the study design. Also, please cite more references to ensure the reliability and the integrity of evidence in the study design that the authors have built and the methodology that they decided to apply to this study.
7. Results: This section would benefit from presenting some headings of subsections according to the analytical methods the authors performed.
8. Discussion: I would like the authors to reorganize this section by incorporating some crucial elements necessary for discussion from the conclusion section. Starting with the summary of the results, the authors need to develop discussion on the potential of this study complementing as the extension of the previous work and current understanding, the implication of the findings of this study, how this study could facilitate future research, the ultimate goal, the challenge, the knowledge and the technology necessary to achieve this goal, the statement about this field in general, and finally the importance of this line of research. Those elements should be written as a possible solution to the weakness and the limitation of this study. I would also ask the authors to include paragraphs of limitations and future directions before the end of the manuscript, in which authors can describe in detail and report all the technical issues that could be brought to the surface.
9. Conclusion: I believe this section would benefit from a single paragraph describing more thoughtful as well as in-depth considerations by the authors. The authors should make their effort to present the take-home message as experts, explaining the theoretical implication as well as the translational application of their research. I believe that it would be necessary to discuss theoretical and methodological avenues in need of refinement, as well as suggestions of a path forward in understanding terpenes activity in the central nervous system. So, many elements discussed in this section can be incorporated into the discussion section.
10. References: Please follow the guidelines of the journal for citation and reference styles (https://www.mdpi.com/journal/brainsci/instructions) and correct them accordingly. The cited papers are to be numbered in the bracket “[]”. The references should provide the abbreviated journal name in italics, the year of publication in bold, the volume number in italics for all the references. Also, please provide doi numbers. I expect 60-70 references for the paper like this.
Overall, the manuscript contains no figure, four tables, and 41 references. The manuscript may carry important value in studying the components of learning capacities which benefit from telerehabilitation in children with specific learning disorders.
I declare no conflict of interest regarding this manuscript.
Best regards,
Reviewer
Author Response
We thank the reviewer for the comments and suggestions that help us to improve the manuscript.
Please consider the following comments:
- Title: Please present the self-explanatory title stating the most significant results of this study.
The title has been revised with a greater focus on the significant results
- methods, the results, and the conclusion. The background should include the general background (one to two sentences), the specific background (two to three sentences), and current issue addressed to this study (one sentence). Especially, the passage “The analysis of some of the variables that may or may not affect 11 the effectiveness of the route is being studied.” reads obscure. I would like the authors to clarify it in the specific background and in the current issue addressed to this study. The result should close with one to two sentences which put the result into a more general context. The conclusion should include one sentence describing the main message using such words like “Here we show”, the potential and the advance this article has provided in the field, and finally a broader perspective (two to three sentences) readily comprehensible to a scientist in any discipline. So, I would like the authors to elaborate the abstract accordingly.
The abstract structure has been revised according to the points suggested.
- Keywords: According to the Journal’s guidelines, please list ten pertinent keywords, which I suggest being specific to the article, yet reasonably common within the subject discipline.Also, please as many as possible in the title and in the first two sentences of the abstract
Specific keywords have been added linked to the topic of the paper and to the title and abstract
- A graphical abstract that will visually summarize the main findings of the manuscript is highly recommended.
We are sorry, we found it difficult to present a graphical abstract on this argument
- Introduction: I believe that this section would benefit from reorganizing this section by focusing the background (general, specific, and the issue addressed to this study) and providing sufficient information on the main constructs of this topic, leading to the objectives. Those constructs should be acknowledgeable to any reader in any discipline and should make this section persuasive enough to put forward the main purpose of current research the authors conduct and the specific purpose the authors have intended by this paper. Those main structures should be organized in a logical and cohesive manner.
I think that more organized and detailed information on specific learning disorders would provide a suitable background here. Thus, I suggest the authors to make their effort to provide a brief overview of the pertinent literature that offer a perspective on definition, causes and symptoms of specific learning disorder, because as it stands, this information is not highlighted in the text. The background should be presented in the following order: schizophrenia in general including brief descriptions of epidemiology, pathogenesis, symptoms, current treatment, and challenge in treatment, comorbidity in specific learning disorders such as attention deficit hyperactivity disorder, conduct disorder, anxiety, and depression. In this regard, I would suggest providing a general overview of pathogenesis and biochemical hallmarks of specific learning disorders, for example ‘mitochondrial impairment as a common motif in neuropsychiatric presentation’ and ‘modelling the neurodevelopmental pathogenesis in neuropsychiatric disorders’. Moreover, I would recommend adding more information on neural substrates of specific learning disorders, specifically on frontal lobe dysfunction, and on related effects on patients’ memory and learning impairments, as ‘Functional interplay between central and autonomic nervous systems’ in this disorder: this information may provide a better understanding of prefrontal cortex’s key role and how its disrupted function may contribute to irregular behavioral responses (doi:10.17219/acem/139572) and therefore to the development of many cognitive dysfunctions (i.e., impaired memory, attention, working memory, problem solving, processing speed, and social cognition) that are common in schizophrenia (https://doi.org/10.3389/fnbeh.2022.998714).
The structure of the whole section has been reorganized and more information has been added about pathogenesis, comorbidities, clinical features, and treatments of learning disabilities.
- Methods:I would like the authors to open this section with an introductory paragraph to help a reader understand the flow of the study design. Also, please cite more references to ensure the reliability and the integrity of evidence in the study design that the authors have built and the methodology that they decided to apply to this study.
An introductory paragraph has been added with more specific information about the study design and methodology.
- Results: This section would benefit from presenting some headings of subsections according to the analytical methods the authors performed.
Headings for each subsection have been added.
- Discussion: I would like the authors to reorganize this section by incorporating some crucial elements necessary for discussion from the conclusion section. Starting with the summary of the results, the authors need to develop discussion on the potential of this study complementing as the extension of the previous work and current understanding, the implication of the findings of this study, how this study could facilitate future research, the ultimate goal, the challenge, the knowledge and the technology necessary to achieve this goal, the statement about this field in general, and finally the importance of this line of research. Those elements should be written as a possible solution to the weakness and the limitation of this study. I would also ask the authors to include paragraphs of limitations and future directions before the end of the manuscript, in which authors can describe in detail and report all the technical issues that could be brought to the surface.
The structure of the whole section has been reorganized and more information has been added about how the present study completes the previous literature, the clinic, and research implications. Furthermore, a ‘limitations and future directions paragraph’ has been added.
- Conclusion: I believe this section would benefit from a single paragraph describing more thoughtful as well as in-depth considerations by the authors. The authors should make their effort to present the take-home message as experts, explaining the theoretical implication as well as the translational application of their research. I believe that it would be necessary to discuss theoretical and methodological avenues in need of refinement, as well as suggestions of a path forward in understanding terpenes activity in the central nervous system. So, many elements discussed in this section can be incorporated into the discussion section.
A single paragraph with the take-home message has been written for the conclusion
- References: Please follow the guidelines of the journal for citation and reference styles (https://www.mdpi.com/journal/brainsci/instructions) and correct them accordingly. The cited papers are to be numbered in the bracket “[]”. The references should provide the abbreviated journal name in italics, the year of publication in bold, the volume number in italics for all the references. Also, please provide doi numbers. I expect 60-70 references for the paper like this.
The references section has been adjusted according to the MDPI reference style and a total number of 66 references are now inserted in the manuscript.
Reviewer 3 Report
1. Abstract needs to be rewritten well. It should be based on the objective / scope of the research, the research method, and the main conclusions. It is necessary to highlight the main findings of the study.
2. There is lack of clarity of ideas in introduction. No sentence consistency or cohesion is found. Ideas are inter-mingled, no connectivity in sentences can be seen. Overall introduction is not well written and needs serious attention.
3. The authors are advised to highlight the inter linkages and should clearly justify the gap of the study.
4. Objectives and significance of the study have serious issue of sentence clarity. There isn't proper justification of outcome variable as well.
5. Please justify why and how the final sample was being selected?
6. tables 3 and 4 need to be improved for anova eta effect size is needed
7. The conclusion can be improved in terms of research gap, contribution and future research.
Learning disorders come in many shapes and forms but this study seem to focus on dyslexia.
SLD impacts roughly 5-10% of school-aged children, making it one of the most common learning disorders in the United States. Those diagnosed with SLD can experience difficulty with reading comprehension, grammar and spelling, and writing. please use dsm5 criteria and definition.
In discussion consider adding that for SLD, the primary area of difficulty is typically in decoding words and understanding concepts. Difficulty with reading can lead to problems in other academic areas such as math, science, and social studies. Those with SLD may also have difficulties with handwriting and organization. Additionally, many of those with SLD may experience difficulty with organization, time management, and setting appropriate goals for themselves. These can be done remotely that is why tele rehab is suitable.
I agree with authors - There has been an emerging shift towards tele rehabilitation as a means of treating people with learning disorders. As research has evolved over the years, evidence has become increasingly available showing just how tele rehabilitation can improve the quality of life of those with learning disabilities. Whether it is speech therapy, occupational therapy or psychological interventions, tele rehabilitation has been seen as an effective way in which to provide treatment for those with learning disabilities.
Author Response
We thank the reviewer for the comments and suggestions that help us to improve the manuscript.
Abstract needs to be rewritten well. It should be based on the objective / scope of the research, the research method, and the main conclusions. It is necessary to highlight the main findings of the study.
The abstract structure has been revised according to the points suggested.
- There is lack of clarity of ideas in introduction. No sentence consistency or cohesion is found. Ideas are inter-mingled, no connectivity in sentences can be seen. Overall introduction is not well written and needs serious attention.
The introduction section has been modified to provide a more exhaustive and cohesive view of the literature and highlight links between the literature and our research study.
- The authors are advised to highlight the inter linkages and should clearly justify the gap of the study.
The gap in the literature to explain our study’s objectives has been better justified.
- Objectives and significance of the study have serious issue of sentence clarity. There isn't proper justification of outcome variable as well.
The objectives and significance of the study have been better explained in the text.
- Please justify why and how the final sample was being selected?
A more detailed description of the inclusion criteria and study procedure has been added.
- tables 3 and 4 need to be improved for anova eta effect size is needed
All eta effect sizes have been added in the text and in tables.
- The conclusion can be improved in terms of research gap, contribution and future research.
The conclusion and discussion section have been improved with a greater focus on the research gap and implications and a new section about ‘Limit and future direction’ has been added.
Learning disorders come in many shapes and forms but this study seem to focus on dyslexia.
SLD impacts roughly 5-10% of school-aged children, making it one of the most common learning disorders in the United States. Those diagnosed with SLD can experience difficulty with reading comprehension, grammar and spelling, and writing. please use dsm5 criteria and definition.
The DSM-5 criteria have been added in the introduction section.
In discussion consider adding that for SLD, the primary area of difficulty is typically in decoding words and understanding concepts. Difficulty with reading can lead to problems in other academic areas such as math, science, and social studies. Those with SLD may also have difficulties with handwriting and organization. Additionally, many of those with SLD may experience difficulty with organization, time management, and setting appropriate goals for themselves. These can be done remotely that is why tele rehab is suitable.
I agree with authors - There has been an emerging shift towards tele rehabilitation as a means of treating people with learning disorders. As research has evolved over the years, evidence has become increasingly available showing just how tele rehabilitation can improve the quality of life of those with learning disabilities. Whether it is speech therapy, occupational therapy or psychological interventions, tele rehabilitation has been seen as an effective way in which to provide treatment for those with learning disabilities.
We have added these relevant aspects to the manuscript.
Round 2
Reviewer 2 Report
27 February 2023
Manuscript ID: brainsci-2243713
Type: Review
Title: ‘Effectiveness of telerehabilitation pathways in Specific Learning Disorders’ by Capodieci A et al., submitted to Brain Sciences
Dear Authors,
I am pleased to see that the authors have tried to solve the issues I raised in the previous round of the peer review session.
Telerehabilitation applies information and communication technologies to provide rehabilitation services to people who reside remotely. In the present research article entitled ‘Effectiveness of telerehabilitation pathways in Specific Learning Disorders’, Capodieci and colleagues investigated the components of learning capacities which benefit from telerehabilitation in children with specific learning disorders.
The main strength of this manuscript is that it addresses an interesting and timely question, providing a captivating interpretation addressing the improvement of reading fluency and accuracy and text dictation in the training group, compared to those on the waiting list.
In general, I think the idea of this article is really interesting and the authors’ fascinating observations on this timely topic may be of interest to the readership of Brain Sciences. However, some comments, as well as some crucial evidence that should be included to support the author’s argumentation, needed to be addressed to improve the quality, the adequacy, and the readability of the manuscript prior to the publication. My overall opinion is to publish this paper after the authors have carefully considered my suggestions below, in particular reshaping parts of the ‘Introduction’ and ‘Discussion’ sections by adding more evidence.
Please consider the following comments:
1. Introduction: I believe that this section would benefit from reorganizing this section by focusing the background (general, specific, and the issue addressed to this study) and providing sufficient information on the main constructs of this topic, leading to the objectives. Those constructs should be acknowledgeable to any reader in any discipline and should make this section persuasive enough to put forward the main purpose of current research the authors conduct and the specific purpose the authors have intended by this paper. Those main structures should be organized in a logical and cohesive manner.
I think that more organized and detailed information on specific learning disorders would provide a suitable background here. Thus, I suggest the authors to make their effort to provide a brief overview of the pertinent literature that offer a perspective on definition, causes and symptoms of specific learning disorder, because as it stands, this information is not highlighted in the text. The background should be presented in the following order: schizophrenia in general including brief descriptions of epidemiology, pathogenesis, symptoms, current treatment, and challenge in treatment, comorbidity in specific learning disorders such as attention deficit hyperactivity disorder, conduct disorder, anxiety, and depression. In this regard, I would suggest providing a general overview of pathogenesis and biochemical hallmarks of specific learning disorders, for example ‘mitochondrial impairment as a common motif in neuropsychiatric presentation’ and ‘modelling the neurodevelopmental pathogenesis in neuropsychiatric disorders’. Moreover, I would recommend adding more information on neural substrates of specific learning disorders, specifically on frontal lobe dysfunction, and on related effects on patients’ memory and learning impairments, as ‘Functional interplay between central and autonomic nervous systems’ in this disorder: this information may provide a better understanding of prefrontal cortex’s key role and how its disrupted function may contribute to irregular behavioral responses (doi:10.17219/acem/139572) and therefore to the development of many cognitive dysfunctions (i.e., impaired memory, attention, working memory, problem solving, processing speed, and social cognition) that are common in schizophrenia (https://doi.org/10.3389/fnbeh.2022.998714).
2. Discussion: Main elements are presented in this section; however, I would like the authors to abridge it to about 1500 words, focusing on not loosing the quality of this section and on reorganize it in a way that a reader comprehends it easily.
3. I recommend presenting figures in color.
Overall, the manuscript contains no figure, four tables, and 66 references. The manuscript may carry important value in studying the components of learning capacities which benefit from telerehabilitation in children with specific learning disorders. I hope that, after careful revisions, this paper can meet the Journal’s high standards for publication. I am available for a new round of revision of this paper.
I declare no conflict of interest regarding this manuscript.
Best regards,
Reviewer
Author Response
Please consider the following comments:
- Introduction: I believe that this section would benefit from reorganizing this section by focusing the background (general, specific, and the issue addressed to this study) and providing sufficient information on the main constructs of this topic, leading to the objectives. Those constructs should be acknowledgeable to any reader in any discipline and should make this section persuasive enough to put forward the main purpose of current research the authors conduct and the specific purpose the authors have intended by this paper. Those main structures should be organized in a logical and cohesive manner.
The introduction is now organized in a coherent way and a lot of information has been included with respect to the present issue so that readers from other areas can also understand the topic and objectives of the study.
I think that more organized and detailed information on specific learning disorders would provide a suitable background here. Thus, I suggest the authors to make their effort to provide a brief overview of the pertinent literature that offer a perspective on definition, causes and symptoms of specific learning disorder, because as it stands, this information is not highlighted in the text. The background should be presented in the following order: schizophrenia in general including brief descriptions of epidemiology, pathogenesis, symptoms, current treatment, and challenge in treatment, comorbidity in specific learning disorders such as attention deficit hyperactivity disorder, conduct disorder, anxiety, and depression. In this regard, I would suggest providing a general overview of pathogenesis and biochemical hallmarks of specific learning disorders, for example ‘mitochondrial impairment as a common motif in neuropsychiatric presentation’ and ‘modelling the neurodevelopmental pathogenesis in neuropsychiatric disorders’. Moreover, I would recommend adding more information on neural substrates of specific learning disorders, specifically on frontal lobe dysfunction, and on related effects on patients’ memory and learning impairments, as ‘Functional interplay between central and autonomic nervous systems’ in this disorder: this information may provide a better understanding of prefrontal cortex’s key role and how its disrupted function may contribute to irregular behavioral responses (doi:10.17219/acem/139572) and therefore to the development of many cognitive dysfunctions (i.e., impaired memory, attention, working memory, problem solving, processing speed, and social cognition) that are common in schizophrenia (https://doi.org/10.3389/fnbeh.2022.998714).
Much information has been added regarding SLD, including definition, causes, symptoms, most effective treatments according to the literature, comorbidities, and also information regarding biological aspects and neurological substrates.
- Discussion: Main elements are presented in this section; however, I would like the authors to abridge it to about 1500 words, focusing on not loosing the quality of this section and on reorganize it in a way that a reader comprehends it easily.
The discussion is 1200 words long and has been rearranged to make it more readable
- I recommend presenting figures in color.
There are no figures in the text
Reviewer 3 Report
thank you for addressing my concerns
Author Response
Thank you for your suggestion to improve the manuscript